# GLIS1, Correlated with Immune Infiltrates, Is a Potential Prognostic Biomarker in Prostate Cancer

**DOI:** 10.3390/ijms25010489

**Published:** 2023-12-29

**Authors:** Qiang Peng, Tingting Xie, Yuliang Wang, Vincy Wing-Sze Ho, Jeremy Yuen-Chun Teoh, Peter Ka-Fung Chiu, Chi-Fai Ng

**Affiliations:** SH Ho Urology Centre, Department of Surgery, The Chinese University of Hong Kong, Hong Kong, China; pengqiang@surgery.cuhk.edu.hk (Q.P.); txie@surgery.cuhk.edu.hk (T.X.); ylwang@surgery.cuhk.edu.hk (Y.W.); vincyho@surgery.cuhk.edu.hk (V.W.-S.H.); jeremyteoh@surgery.cuhk.edu.hk (J.Y.-C.T.)

**Keywords:** GLIS1, prostate cancer, prostate adenocarcinoma, infiltrating immune cell, prognosis, lymphocytes, biomarker

## Abstract

Prostate cancer (PCa) is a prevalent malignant disease and the primary reason for cancer-related mortality among men globally. GLIS1 (GLIS family zinc finger 1) is a key regulator in various pathologies. However, the expression pattern, clinical relevance, and immunomodulatory function of GLIS1 in PCa remain unclear. In this study, GLIS1 was discovered to serve as a key gene in PCa by integrating mRNA and miRNA expression profiles from GEO database. We systematically explored the expression and prognostic values of GLIS1 in cancers using multiple databases. Additionally, we examined the functions of GLIS1 and the relationship between GLIS1 expression levels and immune infiltration in PCa. Results showed that GLIS1 was differentially expressed between normal and tumor tissues in various cancer types and was significantly low-expressed in PCa. Low GLIS1 expression was associated with poor PCa prognosis. GLIS1 was also involved in the activation, proliferation, differentiation, and migration of immune cells, and its expression showed a positive correlation with the infiltration of various immune cells. Moreover, GLIS1 expression was positively associated with various chemokines/chemokine receptors, indicating the involvement in regulating immune cell migration. In summary, GLIS1 is a potential prognostic biomarker and a therapeutic target to modulate anti-tumor immune response in PCa.

## 1. Introduction

Prostate cancer (PCa) is the second most common cancer in men and ranks fifth among cancer-related mortality in men worldwide [1,2]. The incidence of PCa varies geographically and is more prevalent in developed countries, especially in North America, Europe, and Oceania. In China and other Asian countries, the incidence of this disease is rising, and cancers are first diagnosed in the middle/late stages [1,3]. Because the clinical manifestations of early-stage PCa are difficult to identify, an approximate majority of patients are diagnosed belatedly, leading to increased patient mortality [4]. PCa is highly heterogeneous, complex in composition and pathogenesis, so it is difficult to find suitable and effective therapeutic targets. Current clinical treatment options for PCa include prostatectomy surgery, radiation therapy, and androgen deprivation therapy. These therapies can only temporarily limit the progression of PCa. Cancer can advance to the metastatic stage within 18–24 months, and these patients often have poor prognosis with an average survival of 12 months [5,6]. Immunotherapy is a hopeful approach that has demonstrated anti-tumor benefits in PCa [7]. Tumor cells escape the immune system by binding to CTLA-4 or PD-1, promoting the differentiation and activity of Treg cells and suppressing the growth and activity of effector T cells [8,9]. But in PCa, many patients do not respond to monoclonal antibody therapy targeting PD1 or CTLA-4, possibly because tumor cells achieve immune escape through other pathways [10,11,12]. Therefore, exploring new molecules with immunomodulatory functions to treat PCa is urgent and crucial. Recent studies have shown that tumor-infiltrating immune cells can affect the prognosis of PCa patients and the efficacy of anti-tumor immunotherapy [13,14,15]. However, the molecular immune-related mechanisms of PCa remain ambiguous. Therefore, new molecular biomarkers and therapeutic approaches related to the immune infiltration in PCa are urgently needed to meet the challenging clinical needs.

GLIS1 is a GLI-related Krüppel-like zinc finger protein that acts as a transcription activator and repressor. *GLIS1* is known to play a key role in regulating many physiological processes and has also been widely reported to be linked to various pathologies, including oncogenesis [16]. GLIS1, in conjunction with several other reprogramming factors, markedly increased the efficiency of generating induced pluripotent stem cells (iPSC) from somatic cells, suggesting their roles in cellular differentiation, proliferation, and stem cell renewal [17,18,19]. Regarding tumors, *GLIS1* in cancer cells is involved in cell migration, invasion, and tumorigenesis [20,21]. A study by Charles et al. demonstrated that *GLIS1* is associated with the *WNT* gene expression and epithelial-to-mesenchymal transition (EMT) features in breast cancer cells. Their study showed that co-transfection of *GLIS1* and *CUX1* synergistically stimulated *TCF*/*β-catenin* transcriptional activity, thereby increasing cell migration and invasion [22]. A work conducted by Rong et al. demonstrated that *GLIS1* accelerates the process of CD8+ T cell exhaustion in hepatocellular carcinoma by modulating the *SGK1*-*STAT3*-*PD1* pathway, and exerts a synergistic effect with anti-PD1, providing a prospective method for cancer immunotherapy [23].

Many studies in recent years have revealed the presence of differentially expressed genes (DEGs) and differentially expressed microRNA (DEMs) in PCa [24,25,26]. However, the analysis results obtained by different researchers vary. There are still unsolved concerns about the interaction of DEGs and DEMs during Pca development, and further analysis combining different studies is urgently needed to reveal the real targets.

In this work, we obtained DEGs and DEMs between normal and PCa tissue samples by applying four mRNA profiling datasets (GSE183019, GSE134073 [27], GSE69223 [28], and GSE88808 [29]) and two miRNA profiling datasets (GSE89193 [29] and GSE60117 [30]) from the Gene Expression Omnibus (GEO) database. Subsequently, target genes of total DEMs (tDEM–TGs) were predicted, followed by overlapping analysis between tDEM–TGs and total DEGs (tDEGs) to filter potential key genes and miRNAs. We then comprehensively evaluated the differential expression of key gene *GLIS1* and its prognostic value in PCa through The Cancer Genome Atlas (TCGA) database. Additionally, we performed Gene Ontology (GO) analysis, Gene Set Enrichment Analysis (GSEA), and other methods to investigate the potential roles of *GLIS1* in tumor development and tumor immune microenvironment. This study is the first to comprehensively explore the relationship between *GLIS1* expression and its clinical, molecular, and immunotherapeutic aspects in PCa, shedding us new insights into the crucial role of *GLIS1* in PCa.

## 2. Results

### 2.1. Identification of tDEGs, tDEMs and tDEM-TGs in Prostate Cancer

In order to identify key genes that exhibit differential expression between PCa patients and normal controls, we employed mRNA and miRNA-targeted strategy to help filter them. We selected four mRNA expression profiling datasets (GSE183019, GSE134073, GSE69223, and GSE88808) and two non-coding RNA expression profiling datasets (GSE60117 and GSE89193) from the GEO database. Regarding mRNA, 2304, 5094, 3591, and 887 DEGs were obtained from GSE183019, GSE134073, GSE69223, and GSE88808. UpSet plot provided an efficient way to visualize intersections of those up or down-regulated DEGs in the four datasets, and a total of 198 DEGs (tDEGs) were obtained. Of these, 50 genes were up-regulated and 148 genes were down-regulated (Figure 1A,B). The same operations were carried out on miRNA. The two miRNA datasets shared a total of 18 DEMs (tDEMs), including seven upregulated and 11 downregulated miRNAs (Figure 1C,D). Next, target genes for tDEMs were predicted using TargetScan, miRWalk, miRTarBase, and ENCORI databases. Then, 3836 consistent target genes, termed tDEM–TGs, were found by their intersection using Venn’s analysis (Figure 1E).

### 2.2. GO and Pathway Enrichment Analysis

To obtain further insight into the function of the tDEGs and tDEM–TGs in PRAD, we performed GO and Kyoto Encyclopedia of Genes and Genomes (KEGG) pathway enrichment analysis for the identified tDEGs and tDEM–TGs by using the R package “cluster Profiler”. The top 10 most significantly enriched GO terms are depicted in Appendix A. Results showed that the GO enrichment analysis varied between tDEGs and tDEM–TGs (Appendix A). Biological process (BP) in GO analysis revealed that tDEGs were significantly enriched in muscle tissue-associated part. As for tDEM–TGs, they are mainly responsible for regulating the catabolic and autophagy-related processes. The cell component (CC) analysis of tDEGs revealed a significant enrichment in muscle tissue-related components, and the results of tDEM–TGs focused on focal adhesion, cell-substrate adherens junction, and endocytic membrane transport. Similar disparities were also evident in the molecular function (MF) analysis; tDEGs were mainly enriched in transmembrane transporters and channel activity, but tDEM–TGs were enriched in ubiquitin associated function, such as ubiquitin protein ligase binding and transferase activity. Additionally, the KEGG pathway analysis further uncovered the differences between tDEGs and tDEM–TGs (Appendix A).

### 2.3. Identification of Key Genes and miRNAs

To identify potential key genes, we aligned tDEM–TGs with tDEGs to form an intersection, which were referred to as key genes for further exploration (Figure 1F). Moreover, survival analysis based on the TCGA database was used to confirm the initial identified key genes, and only one of these 42 shared key genes, *GLIS1*, was discovered to be associated with overall survival (OS) of PCa patients. The differential expression of *GLIS1* in different GEO datasets were listed in Table 1. Next, we examined candidate miRNAs implicated in the regulation of GLIS1 from 18 tDEMs and identified two up-regulated key miRNAs and one down-regulated key miRNA (Table 2).

To test the clinical applicability of key miRNAs, we examined their diagnostic significance through receiver operating characteristic (ROC) curve analysis (Appendix A). The AUC values of hsa-miR-663b in GSE60117 and hsa-miR-153 in GSE89193 were 0.943 and 0.875, respectively, which demonstrate a good diagnostic value for discrimination between normal controls and PCa patients.

### 2.4. Diagnostic and Prognostic Values of GLIS1 in Pan-Cancer

To further compare the transcriptional levels of *GLIS1* in different human cancers with those of normal samples, we analyzed the expression levels of *GLIS1* mRNA in various cancer types using the RNA-seq data from TCGA and presented the results in Figure 2A. This analysis revealed that *GLIS1* expression level was significantly downregulated in nearly all tumor types compared to the normal, including prostate adenocarcinoma (PRAD), bladder urothelial carcinoma (BLCA), breast invasive carcinoma (BRCA), cervical squamous cell carcinoma and endocervical adenocarcinoma (CESC), colon adenocarcinoma (COAD), glioblastoma multiforme (GBM), kidney chromophobe (KICH), brain lower grade glioma (LGG), liver hepatocellular carcinoma (LIHC), lung adenocarcinoma (LUAD), ovarian serous cystadenocarcinoma (OV), rectum adenocarcinoma (READ), skin cutaneous melanoma (SKCM), stomach adenocarcinoma (STAD), testicular germ cell tumors (TGCT), thyroid carcinoma (THCA), and uterine corpus endometrial carcinoma (UCEC). In addition, *GLIS1* also showed good discriminating ability in our four datasets, among which the AUC values of *GLIS1* were all greater than 0.80 except for GSE183019, indicating it might be a promising biomarker for PCa diagnosis (Figure 2B).

Survival analysis was then performed to ensure the application prospect of *GLIS1*. The association between *GLIS1* expression and OS was first calculated for 33 tumors in the TCGA database using log-rank tests. As depicted in Figure 1C, the forest plot displayed that *GLIS1* could significantly affect the OS of GBMLGG (lower grade glioma and glioblastoma, *p* = 0.001), LGG (*p* = 0.003), SKCM (*p* = 0.003), SARC (sarcoma, *p* = 0.021), THCA (*p* = 0.043), and PRAD (*p* = 0.017) patients (Figure 2C). These tumors all showed that *GLIS1* was associated with patient outcome, suggesting that *GLIS1* has great potential to serve as a prognostic indicator molecule in pan-cancer.

### 2.5. GLIS1 Was Associated with the Prognosis of Prostate Cancer Patients

The mRNA expression levels of *GLIS1* in PRAD were divided as either low or high expression based on the median value (0.94 for FPKM). Clinicopathological features were detailed in Table 3. The downregulation of *GLIS1* in PCa tissues was validated using the TCGA-PRAD dataset (Figure 3A). The expression of *GLIS1* mRNA in the PCa group was markedly lower than that detected in the adjacent normal group, as evidenced by paired specimens (Figure 3B, *p* < 0.001). ROC and survival analysis were conducted to evaluate the application prospect of *GLIS1* (Figure 3C–E). The ROC curve analysis showed that the expression levels of *GLIS1* mRNA could distinguish PCa tissues from normal tissues, with an AUC value of 0.839 (Figure 3C). The Kaplan–Meier survival curve and log-rank tests for OS were conducted on a cohort of 499 patients who had complete follow-up data. Figure 3D demonstrated that the OS is shorter in PCa patients with low *GLIS1* expression compared to those with high *GLIS1* levels (*p* = 0.016). Furthermore, we conducted a univariate Cox proportional hazards regression analysis and found a significant correlation between low *GLIS1* expression and poorer OS (HR = 0.122, *p* = 0.046) in PCa, indicating that *GLIS1* expression can affect the prognosis of PCa patients. Multivariate analysis showed a trend for *GLIS1* expression as an independent indicator for OS (HR = 0.157, *p* = 0.092) (Table 4). The progression-free survival (PFS) analysis revealed that patients with low *GLIS1* expression (bottom 25%) have significantly shorter PFS than patients with elevated *GLIS1* levels (top 25%) (*p* = 0.012) (Figure 3E). To further explore the contribution of *GLIS1* to the clinical characteristics of PCa, we investigated the association between *GLIS1* expression and clinicopathological characteristics using the Wilcoxon rank sum test and logistic regression analysis. As shown in Figure 3F–J and Table 5, with the increase in tumor T stage, N stage, Gleason score, PSA value, and positive residual tumor, the expression levels of *GLIS1* in PCa were significantly reduced.

### 2.6. Functional Enrichment and Pathway Analysis of High and Low-GLIS1 Expression Samples

The survival impact of *GLIS1* on PCa patients suggests its potential clinical application value. Subsequently, we examined the RNA sequencing data obtained from the TCGA database and divided PCa samples into a high *GLIS1* expression group or low *GLIS1* expression group. Between the samples with high and low *GLIS1* expression, 1011 DEGs were discovered, of which 858 genes were upregulated and 153 were downregulated. The DEGs’s expression is shown in a volcano plot and heatmap (Figure 4A,B). Following this, co-expression functions in PCa patients were predicted using GO enrichment and KEGG pathway analysis. In the BP, the DEGs were significantly clustered into items that associated with muscle tissue system, including “muscle contraction”, “muscle system process”, and “smooth muscle contraction”. The remaining BP terms, including “hormone metabolic process”, “regulation of synapse assembly”, and “second-messenger-mediated signaling”, etc. (Figure 4C). Meanwhile, the DEGs–associated CC were “collagen-containing extracellular matrix” and other muscle tissue related terms, including Z disc, I band, contractile fiber, sarcolemma and sarcomere. “Transmembrane transporter complex” and “intrinsic component of synaptic membrane” are also significant CC terms (Figure 4D). The top 10 significant MF terms were mainly associated with channel activity, including “passive transmembrane transporter activity”, “substrate-specific channel activity”, “ion gated channel activity”, and so on (Figure 4E). For the KEGG analysis, these *GLIS1*–assocaited DEGs were mainly enriched in neuroactive ligand-receptor interaction, cAMP signaling pathway, calcium signaling pathway, focal adhesion, vascular smooth muscle contraction, cGMP-PKG signaling pathway, protein digestion and absorption, pancreatic secretion, chemical carcinogenesis, and drug metabolism. The 10 most significant KEGG pathways were shown in Figure 4F.

Additional investigation using a GSEA function analysis showed that *GLIS1* also took part in several immune-related pathways and influenced many processes of immune cells. These findings showed that *GLIS1* was linked to the following processes: activation of natural killer (NK) cell, T cell, and macrophage (Figure 5A); proliferation of B cell and T cell (Figure 5B); differentiation of T cell, B cell, dendritic cells (DC), and myeloid leukocyte (Figure 5C); and migration of T cell, neutrophil, myeloid leukocyte, and macrophage (Figure 5D).

### 2.7. The Correlation between GLIS1 Expression and Immune Infiltration Levels

Considering the GSEA results found that *GLIS1* was involved in tumor immune regulation, we next performed single-sample Gene Set Enrichment Analysis (ssGSEA) to detect the correlation between *GLIS1* mRNA expression and immune cell infiltration levels in PCa. The association between *GLIS1* mRNA expression and the infiltration of immune cells was shown in Figure 6A. The results indicated that *GLIS1* expression was positively correlated with the infiltration of NK cells, and the Spearman correlation was as high as 0.712 (*p* < 0.01). *GLIS1* expression was also positively associated with the infiltration of mast cells, neutrophils, effector memory T cells (Tem), gamma delta T cells (Tgd), immature dendritic cells (iDC), Th1 cells, DC, B cells, cytotoxic T cells, plasmacytoid dendritic cells (pDC), T cells, macrophages, eosinophils, CD8+ T cells, NK CD56dim cells, T follicular helper cells (TFH), NK CD56bright cells, Th17 cells (all *p*  <  0.01), but was negatively correlated with the infiltration of Th2 cells (*p* < 0.05, Figure 6A). Additionally, we assessed the abundance of key immune cells in the groups with low and high *GLIS1* expression. The results showed that in PCa, 18 major types of immune cells (including NK cells, macrophages, DC, T cells, CD8+ T cells, neutrophils, and B cells, etc.) showed more abundant infiltration in the high *GLIS1* group compared to the low *GLIS1* group (*p* < 0.001). In contrast, the abundance of Th2 cells was lower in the high *GLIS1* group (Figure 6B).

In addition, the analyses by Tumor Immune Estimation Resource (TIMER) software showed that the *GLIS1* mRNA expression exhibited positive correlations with the infiltration of B cells (R = 0.164, *p* < 0.001), CD8+ T cells (R = 0.214, *p* < 0.001), CD4+ T cells (R = 0.335, *p* < 0.001), macrophages (R = 0.469, *p* < 0.001), neutrophils (R = 0.235, *p* < 0.01), and DC (R = 0.297, *p* < 0.001), but had a negative correlation with the purity of tumor (R = −0.499, *p* < 0.001) (Figure 6C). Hence, *GLIS1* may play a critical role in regulating antitumor immunity. Furthermore, we utilized TIMER to investigate the relationship between *GLIS1* expression and various markers associated with immune infiltration in PCa. After the correlation adjustment by purity, the results revealed a significant correlation between *GLIS1* expression levels and most of the marker sets of CD8+ T cell, T cell (general), B cell, monocyte, TAM (tumor-associated macrophage), M1 macrophage, NK cell, DC, Th1 cell, Th2 cell, Treg cell, and T cell exhaustion (Table 6). A somatic copy number alterations (SCNAs) module showed that the deep depletion and arm-level deletion of *GLIS1* were mainly significantly associated with immune cell infiltration levels in PRAD (Appendix A). Collectively, our findings indicate that *GLIS1* may have a significant impact on the infiltration of immune cells in PCa.

### 2.8. The Association between GLIS1, Chemokines and Chemokine Receptors

To further clarify the connection between *GLIS1* and the migration of immune cells, we integrated chemokines and chemokine receptors in Figure 7A–I. The results demonstrated positive correlations between *GLIS1* expression and lymphocyte-associated chemokines and chemokine receptors, including *CX3CL1*, *CX3CR1*, *CXCL1*, *CXCR2*, *CXCL12*, *CXCR4*, *CCL2*, *CCR2*, *CCL5*, *CCR5*, *CCL14*, *CCR5*, *CCL21*, *CCR7*, *CCL28*, *CCR10*, *XCL1*, and *XCR1*. These chemokines and chemokine receptors were upregulated with *GLIS1* expression levels increased. Hence, high *GLIS1* expression may contribute to the migration of immune cells to the tumor tissues.

## 3. Discussion

It is well recognized that miRNA is a crucial molecular tool for non-invasive diagnosis and prognosis of cancer. MiRNAs can bind to the 3′ untranslated region (3′ UTR) of target mRNAs, leading to the degradation and translational repression of mRNAs, exerting regulatory roles in cancer progression or suppression [31,32]. Therefore, it is important to identify the most suitable DEGs that can be used as diagnostic markers or therapeutic targets by analyzing gene expression changes in combination with miRNAs. In this study, we first identified 198 tDEGs and 18 tDEMs between PCa tissues and normal tissues from GEO datasets. Through the utilization of GO and KEGG enrichment analysis, we acquired an in-depth understanding of these genes associated with the initiation and development of PCa. To learn the general patterns of molecular changes in PCa, we then uncovered *GLIS1* by aligning tDEM–TGs with tDEGs. *GLIS1* exhibited reduced expression levels in PCa and demonstrated a significant association with patient survival. The expression of *GLIS1* was found to be downregulated in a cohort of malignancies comparing tumor and normal tissues from independent datasets in the Genotype-Tissue Expression (GTEx) database and TCGA database. Additionally, *GLIS1* participated in the biological process of immune cells entering tumor tissues and improved the tumor microenvironment (TME) of patients, and thus may affect the development of PCa and patients’ prognosis. Therefore, *GLIS1* has great potential to serve as an immune-related biomarker in PCa. Our research provides a more comprehensive understanding of the mechanisms by which *GLIS1* involved in the development of PCa.

The study of tumor microenvironment, including various cell clusters such as immune cells, tumor cells, and fibroblasts, has gained significant attention in recent years and has a profound impact on the progression of cancers [33,34]. Rong et al. demonstrated that *GLIS1* contributes to CD8+ T cell exhaustion by transcriptionally regulating the *SGK1*-*STAT3*-*PD1* signaling pathway in HCC. Suppression of *GLIS1* in CD8+ T cells enhances the effectiveness of anti-PD1 treatment, presenting a promising strategy for HCC immunotherapy [23]. These suggested that *GLIS1* might be involved in immune response. Therefore, how *GLIS1* participates in the TME and influences tumor-infiltrating immune cells in PCa deserves exploration. Using the TIMER database, we identified that *GLIS1* exerted an influence on tumor-infiltrating immune cells in PCa. The expression of *GLIS1* exhibited a notable inverse association with tumor purity and was positively correlated with the infiltration of various immune cells in PCa, including B cells, DC, CD4+ T cells, CD8+ T cells, macrophages, and neutrophils. The ssGSEA methods further demonstrated a positive correlation between *GLIS1* expression and the infiltration of NK cells, NK CD56bright cells, NK CD56dim cells, DC, plasmacytoid DC, immature DC, several types of T cells (Tem, Tgd, Th1 cells, cytotoxic T cells, Th17 cells, CD8+ T cells, and TFH), B cells, mast cells, neutrophils, eosinophils, macrophages, and negatively correlated with Th2 cells. Increased NK cells and CD8+ T cells can strengthen the anti-tumor immune response through the secretion of diverse cytokines and the release of perforin and granzyme [35]. NK cells, which are cytotoxic lymphocytes, consist of two primary subpopulations in humans: CD56bright and CD56dim cells [36]. Both subsets of NK cells enhance their cytolytic ability by generating a diverse range of pro-inflammatory cytokines, including as IFN-γ and TNF-α, demonstrating increased cytotoxicity against malignancies [37]. DC are the most powerful antigen-presenting cells, and their infiltration can facilitate the presentation of tumor-associated or tumor-specific antigens to T cells, thereby initiating primary immune responses [38]. The functions of tumor-associated macrophages act as a “double-edged sword” in tumor development. M1 macrophages play an anti-tumor role by secreting pro-inflammatory factors and chemokines, participating in antigen presentation and immune surveillance, while M2 macrophages promote tumor growth and angiogenesis [39,40]. The increase in B cell infiltration facilitates tumor elimination through its involvement in antibody-dependent cytotoxicity. Th1 cells were reported to inhibit PCa growth by activating CD8+ T cells and NK cell-mediated cytolytic function via the manufacturing of Th1-type cytokines such as IFN-γ and IL-2 [41]. However, Th2 cells are considered to promote tumor progression due to the immunosuppressive substances they release, including TGF-β, IL-4, IL-5, and IL-13 [42]. Research has demonstrated that the infiltrating of Th2 cells is linked to weakened immune response and lower survival time across various cancer types [43,44]. In our study, *GLIS1* negatively regulated the infiltration of Th2 cells, which suggested that the decreased *GLIS1* expression in PCa may help mediate the tumor immune escape. As a result, *GLIS1* is involved in the regulation of TME in PCa by engaging in both cellular and humoral immunity and promoting anti-tumor activities. These findings are exciting, especially considering that there are still many PCa patients who do not respond to anti-PD1 or anti-CTLA-4 therapy, in part because PCa itself has low immune infiltration and in part because tumor cells may achieve immune escape through other pathways. Our study provides a new molecule with immunomodulatory functions to treat PCa. We may first use cells, tissues, and PCa patient-derived xenograft mice models to verify the immune regulatory function of *GLIS1* in PCa and further clarify its molecular mechanism involved in immune regulation. In vitro and in vivo experiments will help us detect the efficacy of *GLIS1* in enhancing PCa immunotherapy, such as the potential synergistic effect in combination with immune checkpoint inhibitors in PCa immunotherapy. The confirmation of in vitro and in vivo experiments will help us conduct human trials to explore the combined effect of GLIS1 monoclonal antibodies with anti-PD1 or anti-CTLA-4 for future PCa treatment.

Concerning GO enrichment analysis, *GLIS1* was found to be involved in several signaling pathways in tumor cells, which is similar to the GO analysis results of tDEGs in PRAD (Figure 4C–E and Appendix A), revealing a high degree of correlation regulation. For example, muscle contraction, and muscle system process for BP, collagen-containing extracellular matrix and contractile fiber for CC, and passive transmembrane transporter activity and channel activity for MF. Furthermore, the GSEA on GLIS-related immune function showed that *GLIS1* affected the activation, proliferation, differentiation, and migration of immune cells. It suggests that *GLIS1* enhances the immunological response of PCa through multiple pathways. Chemokines regulate the positioning and motility patterns of immune cells and are essential for immune cell migration and TME homeostasis. Subsequently, we combined chemokines and chemokine receptors and examined the correlation between *GLIS1* and the migration of immune cells to investigate the potential immunological-related mechanism of *GLIS1* in PCa. The interaction between *CX3CL1* and *CX3CR1* is involved in the recruitment of NK cells, T cells, and monocytes, and is associated with the activation of cytotoxic T lymphocytes and NK cells [45]. *CCL5*/*CCR5* plays a role in the migration of macrophage and NK cells, as well as in the interactions between T cells and DC cells [46]. *CXCL1*/*CXCR2* and *CXCL12*/*CXCR4* are related to neutrophil recruitment and NK cell migration, respectively [47,48]. *CCL21*/*CCR7* plays important roles in T cell recruitment to lymph nodes and in T cell migration [49]. *XCL1*/*XCR1* interaction is associated with NK cell recruitment [50]. In this study, *GLIS1* was positively associated with these chemokines/chemokine receptors, indicating that *GLIS1* may enhance tumor immune infiltration levels via modulating the migration of immune cells in PCa.

Though this study enhances our comprehension of the association between *GLIS1* and PCa, it does have several limitations. Firstly, it is necessary to conduct molecular investigations to validate the mechanisms of *GLIS1* and its influence on the clinical outcome of PCa. Furthermore, critical signaling pathways related to *GLIS1* need to be validated. In addition, the mechanism by which *GLIS1* regulates immune cell infiltration remains to be studied. It is also crucial to incorporate and clarify the connection between *GLIS1* and chemokines/chemokine receptors, which can enhance our comprehension about TME, specifically the immune microenvironment in PCa.

In conclusion, this is the first comprehensive study to elaborate the association between the expression of *GLIS1* and its clinical, molecular, and immunotherapeutic characteristics in PCa, providing us new insights into the important role of *GLIS1* in PCa. Our work has found that the expression of *GLIS1* varies between normal tissue and tumor tissues in various cancer types and is significantly downregulated in PCa, and decreased *GLIS1* expression is associated with unfavorable prognosis in PCa. *GLIS1* also plays a crucial role in the microenvironment of PCa by controlling the infiltration of immune cells into tumors, indicating *GLIS1* as a potential therapeutic target to regulate immune responses against tumors. Consequently, *GLIS1* shows considerable promise as a helpful biomarker and therapeutic target for PCa prognosis prediction and treatment.

## 4. Materials and Methods

### 4.1. GEO Data Extraction and DEGs/DEMs/tDEGs/tDEMs Analysis

The gene expression profiles were obtained from the GEO database and the selection criteria were as stated: (1) prostate adenocarcinoma, (2) time period from 1 January 2015 to 31 December 2022, (3) tissue specimen, (4) homo sapiens, (5) sample count surpasses 30, (6) mRNA or miRNA expression data accessible. Consequently, a total of six datasets were selected, namely GSE183019, GSE134073, GSE69223, GSE88808, GSE60117, and GSE89193.

The DEGs and DEMs between normal and PCa tissue samples in each dataset were examined using “limma” package in R. Subsequently, the cut-off criteria for significant DEGs were defined as a *p* value < 0.05 and a |fold change| > 1.5. Similarly, for significant DEMs, the criteria were set as a *p* value < 0.05 and a |fold change| > 1.2. Volcano plots were subsequently utilized to promptly detect differences in DEGs and DEMs. Additionally, UpSet/Venn plots were employed to obtain the overlapping upregulated or downregulated DEGs/DEMs across all mRNA/miRNA datasets, respectively. The obtained datasets were referred to as tDEGs/tDEMs by combining the commonly shared up-regulated and down-regulated DEGs/DEMs in this investigation.

### 4.2. Enrichment Analysis and Key Genes Acquisition

tDEM–TGs were predicted using the online databases miRTarBase (http://miRTarBase.cuhk.edu.cn/, accessed on 22 May 2023), miRWalk (http://mirwalk.umm.uni-heidelberg.de/, accessed on 22 May 2023), TargetScan (https://www.targetscan.org/, accessed on 25 May 2023), and ENCORI (https://starbase.sysu.edu.cn/, accessed on 25 May 2023) [51,52,53]. The R package “clusterProfiler” (version 4.10.0) was used to analyze GO and KEGG for functional annotation and pathway enrichment analysis of tDEM–TGs and tDEGs. A *p*-value < 0.05 is deemed to be statistically significant. Then, the tDEM–TGs were matched with tDEGs to identify the intersection genes referred to as “key genes” for further investigation.

### 4.3. Expression and Prognosis of GLIS1 in Pan-Cancer

The TCGA database (https://portal.gdc.cancer.gov/) and the GTEx database (https://gtexportal.org/, accessed on 20 June 2023) provided the gene expression matrix and clinical data for each tumor and normal sample. Utilizing the Wilcoxon rank-sum test, differences in *GLIS1* expression levels between tumor and normal tissues were examined. Histogram was drawn busing the R package “ggplot2” (version 3.4.2). The Kaplan–Meier method was employed to evaluate survival with varying levels of *GLIS1* expression. Based on the median value of gene expressions, *GLIS1* in tumor tissues were classified into high and low *GLIS1* expression groups, and the log-rank test analysis was done using “survival” package (version 3.3.1) in R. Visualization was demonstrated by a forest plot.

### 4.4. Analysis of DEGs between the High and Low GLIS1 Expression Groups in PRAD Patients

The RNA sequencing data and clinical information of 499 patients diagnosed with PRAD were obtained through the TCGA database. Wilcoxon rank-sum test was employed to investigate the changes in *GLIS1* expression among PRAD patients with different T stage, N stage, Gleason score, and other clinicopathological features. PRAD Patients were categorized into two groups, low-expression and high-expression, based on their median *GLIS1* expression levels. DEGs were identified by comparing the expression profiles (HTSeq-FPKM) between the high *GLIS1* group and low *GLIS1* group using the “DESeq2” (version 1.34.0) package in R [54]. A |log2 fold change| ≥ 1 and adj. *p*-value < 0.05 are set as the cutoff values for the DEGs. To understand the underlying biological roles and pathways of *GLIS1*-related DEGs, KEGG pathway, and GO analysis focusing on BP, MF, and CC, were analyzed using the “clusterProfiler” R package (version 4.10.0) to determine the functions of the DEGs. In addition, we integrated the GO enrichment data with expression data (log2 fold change) and utilized the R package “GOplot” to compute the corresponding Z-score. Pathways and GO terms with an adj. *p*-value < 0.05 are considered significant. R package “ggplot2” (version 3.3.5) was used for GO and pathway visualization.

### 4.5. Gene Set Enrichment Analysis (GSEA)

GSEA is an analytical tool determining whether a previously identified gene set shows statistically significant, concordant differences between two phenotypes [55]. This study employed the R package “clusterProfiler” (version 4.10.0) [56] to perform GSEA to identify significant function and pathway differences between the high-*GLIS1* group and low-*GLIS1* group. The study chose C5: ontology gene sets in the MSigDB Collections as the reference gene collection. An adj. *p*-value < 0.05, false discovery rate (FDR) < 0.25, and an absolute value of normalized enrichment score (|NES|) > 1 are regarded as significant enrichment.

### 4.6. Immune Infiltration in Tumor Tissues

We utilized the ssGSEA method from the “GSVA” package (version: 1.50.0) in R [57] to measure the abundance of tumor-infiltrating immune cells based on PRAD mRNA FPKM data. The ssGSEA marker genes were obtained from the study conducted by Bindea et al. [58], which encompassed 24 different types of immune cells. The Wilcoxon rank-sum test and Pearson correlation were used to assess the relationship between immune cell infiltration and different *GLIS1* expression groups.

TIMER [59] is a comprehensive online platform used to assess tumor-infiltrating immune cells in various cancer types. The web server can be accessed at https://cistrome.shinyapps.io/timer/, accessed on 1 August 2023. The TIMER encompasses over 10,000 samples across multiple cancer types in the TCGA database. It utilizes a partial deconvolution linear least square regression approach to determine the abundance of immune infiltrates. We assessed the association between *GLIS1* expression and immune infiltrates in PRAD samples, including CD8+ T cells, CD4+ T cells, DC, neutrophils, B cells, and macrophages.

### 4.7. Statistical Analyses

The gene expression HTSeq RNA-seq FPKM value were transformed into log2 (FPKM + 1). The Shapiro–Wilk test was used to test data normality. Non-normally distributed two independent groups were analyzed using the Wilcoxon signed-rank test (when paired) or Mann–Whitney U test (when unpaired). The Mann–Whitney U test was employed to evaluate the expression of *GLIS1* in non-paired samples, whereas the Wilcoxon signed-rank test was utilized for paired samples. The ROC curve was constructed to assess the diagnostic efficacy of *GLIS1* and key miRNAs in the GEO datasets using the “pROC” package in R. The Wilcoxon signed-rank test and Chi-square test were used to analyze the relationship between *GLIS1* expression and clinicopathological characteristics in PCa. The Kaplan–Meier method was utilized to create survival curves, and the log-rank test was employed to evaluate the survival disparities between groups. Univariate and multivariate analyses using Cox proportional hazard modelling were utilized to evaluate the impact of clinical factors on survival. A hazard ratio (HR) > 1 indicates an elevated risk, whereas a HR < 1 indicates a reduced risk. A *p* < 0.05 (two-sided) is considered statistically significant. Statistical analyses were conducted using R (version 3.6.1), SPSS (version 24.0, IBM, New York, NY, USA), or GraphPad Prism (version 8.0, GraphPad Software, La Jolla, CA, USA).

## Figures and Tables

**Figure 1 ijms-25-00489-f001:**
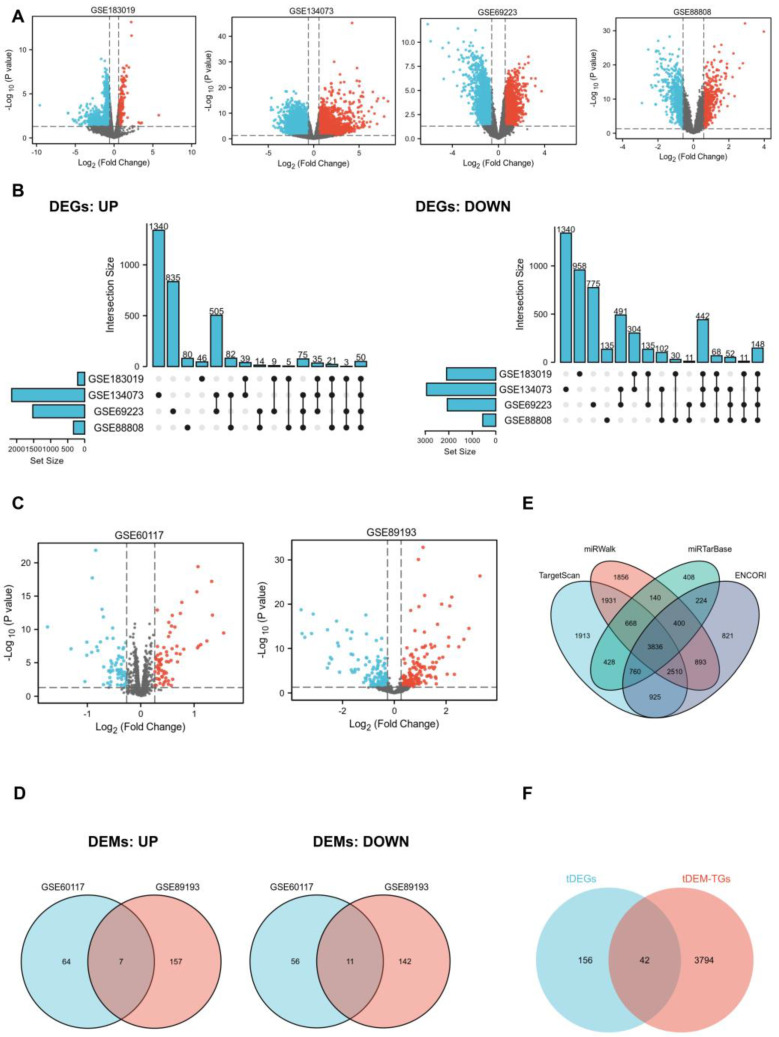
Identification of DEGs, DEMs and key genes. (**A**), Volcano plot of DEGs from GEO datasets GSE183019, GSE134073, GSE69223, and GSE88808, respectively. Upregulated mRNAs are indicated in red, downregulated mRNAs are indicated in blue, and not-significant mRNAs are represented in grey. (**B**), UpSet plot of upregulated (**left**) and down-regulated (**right**) DEGs set from GSE183019, GSE134073, GSE69223, and GSE88808 datasets. (**C**), Volcano plot of DEMs from GSE60117 dataset and GSE89193 dataset, respectively. (**D**), Venn diagram of upregulated DEMs (**left**) and downregulated DEMs (**right**) set from GSE60117 and GSE89193 datasets; (**E**), Target genes of 18 total DEMs (tDEM–TGs) were predicted by miRTarBase, miRWalk, TargetScan, and ENCORI databases. (**F**), Venn diagram of tDEGs and tDEM–TGs to obtain key genes.

**Figure 2 ijms-25-00489-f002:**
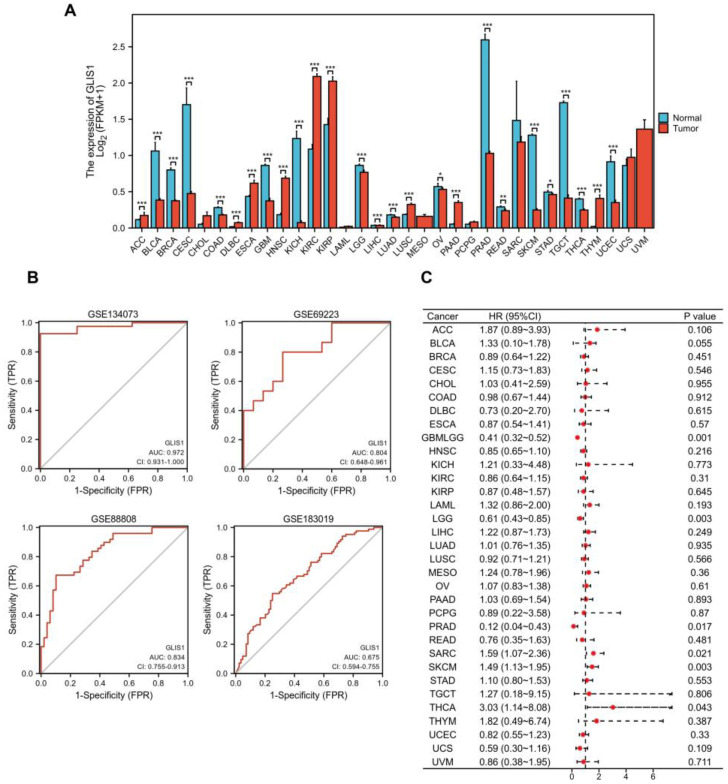
Diagnostic and prognostic values of *GLIS1* in PCa and other cancer types. (**A**), Expression levels of *GLIS1* in 33 tumors were investigated by integrating the normal tissue data from the GTEx database with the tumor tissue data from the TCGA database, * *p* < 0.05, ** *p* < 0.01, *** *p* < 0.001. (**B**), ROC curves of *GLIS1* in GSE183019, GSE134073, GSE69223, and GSE88808. (**C**), Forest plot displayed the correlation between *GLIS1* expression and overall survival time in 33 tumors. The HR estimate is represented graphically by a red dot; the CI is represented by a horizontal dashed line.

**Figure 3 ijms-25-00489-f003:**
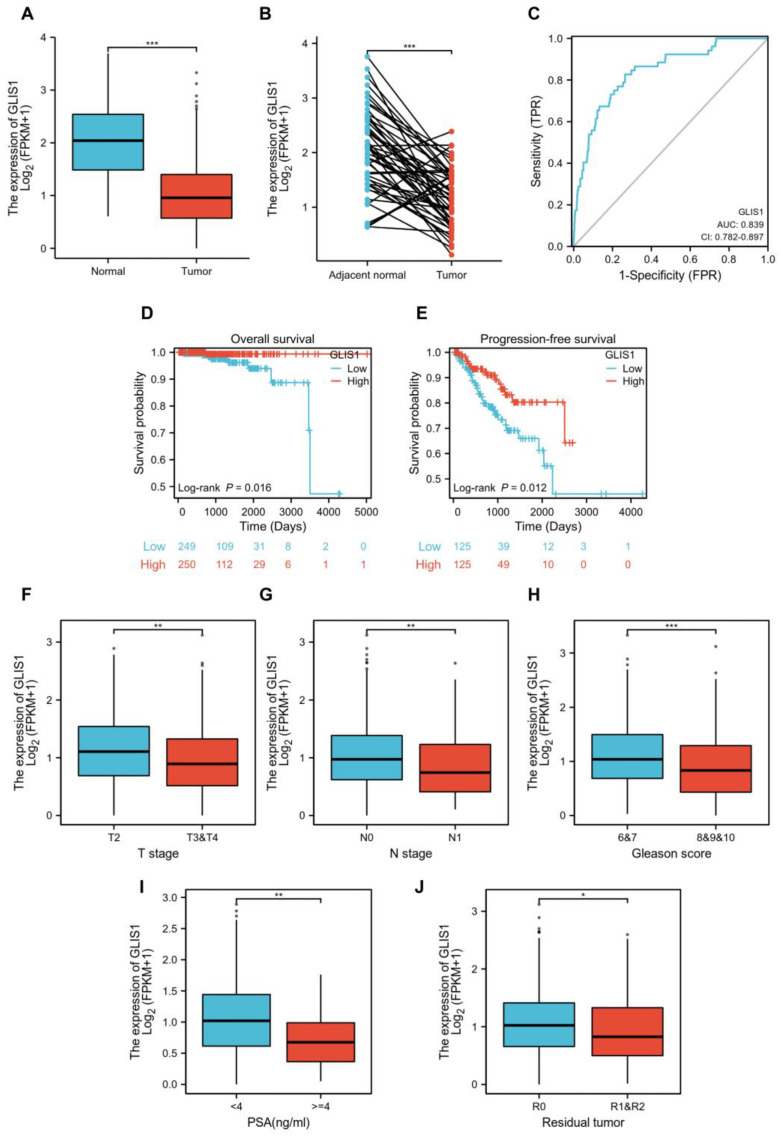
Relationship between *GLIS1* expression and clinicopathological parameters in TCGA-PRAD cohort. (**A**), Expression of *GLIS1* in PRAD and normal tissues. (**B**), Expression of *GLIS1* in PRAD and its paired adjacent tissues. (**C**), ROC analysis of *GLIS1* in PRAD. (**D**), Kaplan–Meier plot and log-rank test analysis of overall survival in PRAD patients with high *GLIS1* expression vs. those with low *GLIS1* expression. (**E**), Kaplan–Meier plot and log-rank test analysis of progression-free survival in PRAD patients with highly expressed *GLIS1* (top 25%) vs. those with lowly expressed *GLIS1* (bottom 25%). (**F**–**J**), Low expression of *GLIS1* was significantly associated with advanced T stage (**F**), lymph node metastasis (**G**), higher Gleason score (**H**), higher PSA value (**I**) and positive residual tumor (**J**). * *p* < 0.05, ** *p* < 0.01, *** *p* < 0.001.

**Figure 4 ijms-25-00489-f004:**
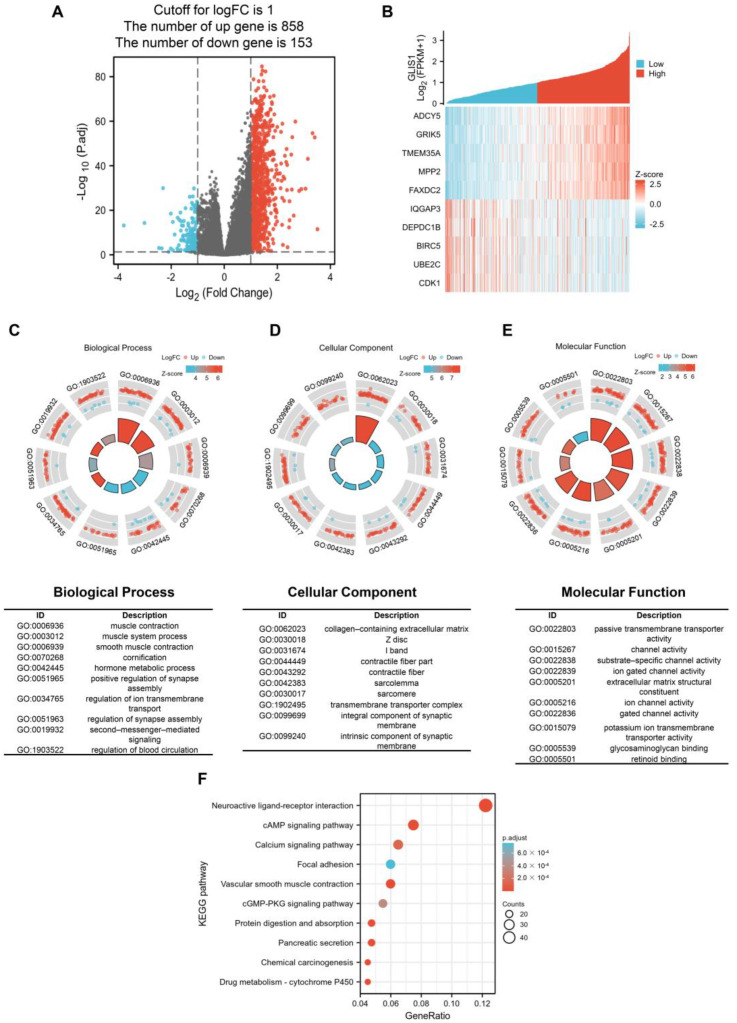
Gene Ontology (GO) and Kyoto Encyclopedia of Genes and Genomes (KEGG) functional enrichment analysis of tumor samples with high *GLIS1* expression vs. those with low *GLIS1* expression. (**A**), Volcano plot of DEGs between tumor samples with high *GLIS1* expression and samples with low *GLIS1* expression. Grey represents not-significant change in gene expression, red represents upregulation and blue represents down-regulation. (**B**), Heatmap of DEGs. (**C**–**E**), GO analysis of DEGs. Detailed information relating to changes in the (**C**) biological process, (**D**) cellular components, (**E**) molecular function. (**F**), KEGG analysis for DEGs.

**Figure 5 ijms-25-00489-f005:**
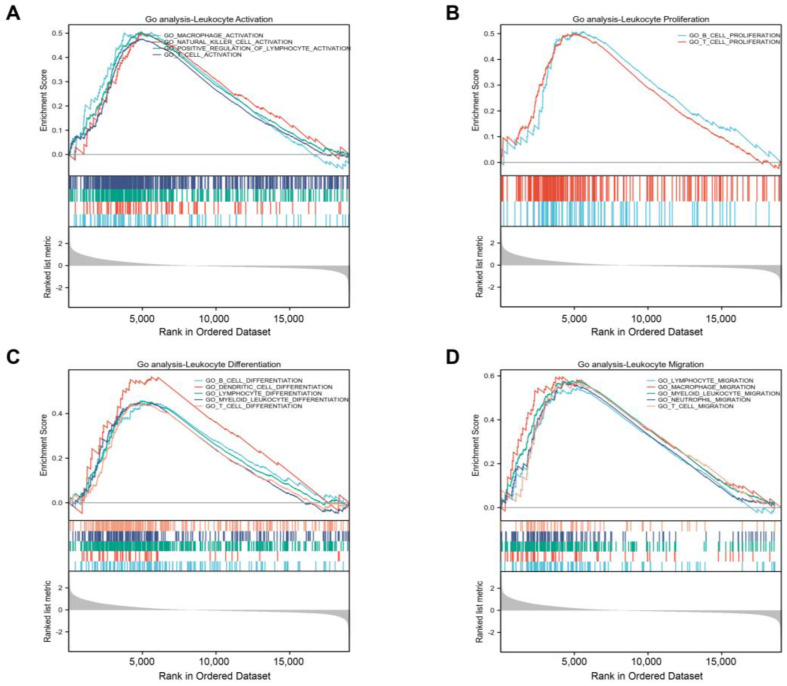
Gene Set Enrichment Analysis (GSEA) identifies immune cell-related pathways. (**A**), *GLIS1* was found to be correlated with the activation of macrophage, natural killer (NK) cell, and T cell. (**B**), *GLIS1* was linked to the proliferation of B cell and T cell. (**C**), *GLIS1* was associated with the differentiation of B cell, dendritic cell (DC), myeloid leukocyte, and T cell. (**D**), *GLIS1* was associated with the migration of macrophage, myeloid leukocyte, neutrophil, and T cell.

**Figure 6 ijms-25-00489-f006:**
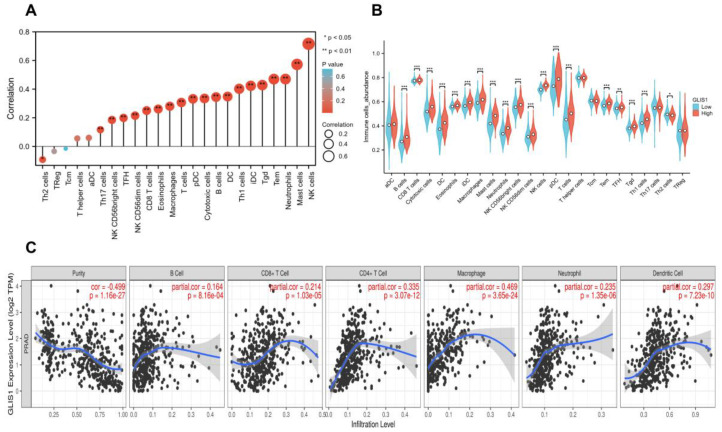
ssGSEA analysis of *GLIS1* and the correlation of *GLIS1* expression with immune infiltration levels in PRAD. (**A**), The relationship between the infiltration of immune cells and the expression of *GLIS1*. The expression of *GLIS1* showed a positive correlation with the infiltration levels of 19 different types of immune cells in PCa and exhibited a negative correlation with Th2 cells. (**B**), Violin plot showed significant differences in the abundance of major immune cells between the high *GLIS1* group and low *GLIS1* group. * *p* < 0.05, ** *p* < 0.01 and *** *p* < 0.001. (**C**), *GLIS1* expression showed a significant negative correlation with tumor purity and a positive correlation with the immune infiltrating levels of B cells, CD8+ T cells, CD4+ T cells, macrophages, neutrophils, and dendritic cells (DC).

**Figure 7 ijms-25-00489-f007:**
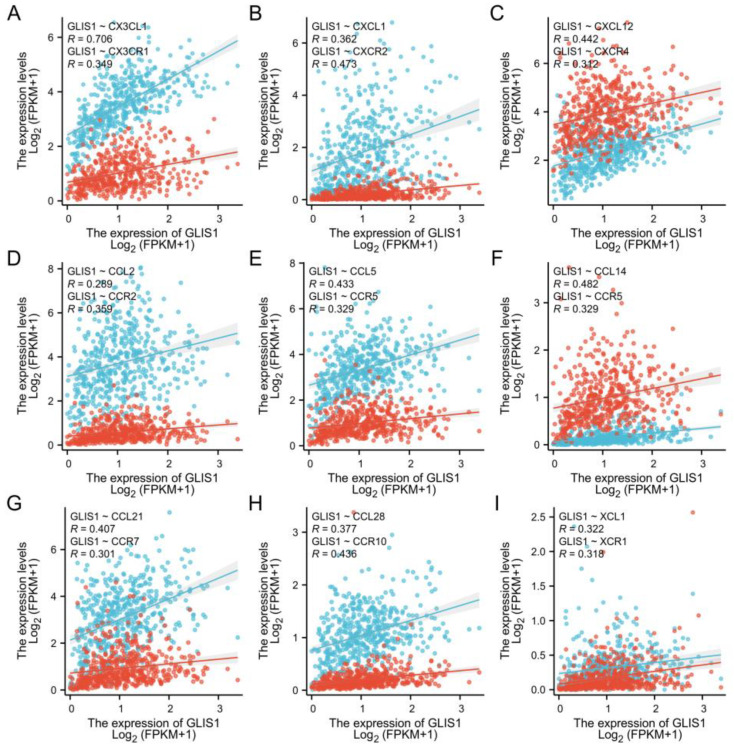
The scatter plot and correlation curve of *GLIS1*, chemokines, and chemokine receptors in PRAD. (**A**–**I**) *GLIS1* was positively associated with *CX3CL1*, *CX3CR1*, *CXCL1*, *CXCR2*, *CXCL12*, *CXCR4*, *CCL2*, *CCR2*, *CCL5*, *CCR5*, *CCL14*, *CCR5*, *CCL21*, *CCR7*, *CCL28*, *CCR10*, *XCL1*, and *XCR1*. *CCL*, CC chemokine ligand; *CCR*, CC chemokine receptor; *CXCL*, CXC chemokine ligand; *CXCR*, CXC chemokine receptor; *XCL*, C chemokine ligand; *XCR*, C chemokine receptor. Dots represent PRAD samples from TCGA, and Spearman’s correlation coefficient (R) was used to evaluate the correlation between *GLIS1* and lymphocyte-associated chemokines (blue) and chemokine receptors (red).

**Table 1 ijms-25-00489-t001:** The differential expression of GLIS1 in different GEO datasets.

Gene	Datasets	LogFC	*p* Value	adj. *p* Value	Expression
*GLIS1*	GSE183019	−0.63	0.000	0.012	DOWN
	GSE134073	−2.09	0.000	0.000	DOWN
	GSE69223	−0.94	0.002	0.017	DOWN
	GSE88808	−0.77	0.000	0.000	DOWN

**Table 2 ijms-25-00489-t002:** The differential expression of candidate miRNAs in different GEO datasets.

Target Gene	miRNA	Datasets	LogFC	*p* Value	adj. *p* Value	Expression
*GLIS1*	hsa-miR-663b	GSE60117	0.28	0.000	0.000	UP
		GSE89193	1.36	0.000	0.000	UP
*GLIS1*	hsa-miR-153	GSE60117	0.35	0.003	0.015	UP
		GSE89193	2.22	0.000	0.000	UP
*GLIS1*	hsa-miR-483-5p	GSE60117	−0.62	-0.010	0.039	DOWN
		GSE89193	−1.10	0.001	0.006	DOWN

**Table 3 ijms-25-00489-t003:** Demographic and clinical characteristics of PRAD patients with low and high GLIS1 expression in TCGA (n = 499).

Clinicopathologic Variables	No. of Cases	GLIS1 Expression Level		χ	*p*
		Low	High		
All cases	499	249(44.3%)	250(55.7%)		
Age				0.019	0.889
≤60	224 (44.9%)	111 (22.2%)	113 (22.6%)		
>60	275 (55.1%)	138 (27.7%)	137 (27.5%)		
Clinical T stage				0.623	0.430
T1 + T2	351 (86.5%)	165 (40.6%)	186 (45.8%)		
T3 + T4	55 (13.5%)	29 (7.1%)	26 (6.4%)		
Pathologic T stage				8.256	0.004
T2	189 (38.4%)	79 (16.1%)	110 (22.4%)		
T3 + T4	303 (61.6%)	167 (33.9%)	136 (27.6%)		
Lymph nodes status				6.477	0.011
Negative	347 (81.5%)	169 (39.7%)	178 (41.8%)		
Positive	79 (18.5%)	51 (12.0%)	28 (6.6%)		
PSA (ng/mL)				7.179	0.007
<4	415 (93.9%)	197 (44.6%)	218 (49.3%)		
≥4	27 (6.1%)	20 (4.5%)	7 (1.6%)		
Gleason score				6.674	0.010
6–7	293 (58.7%)	132 (26.5%)	161 (32.3%)		
8–10	206 (41.3%)	117 (23.4%)	89 (17.8%)		
Survival Status				6.564	0.010
Live	489 (98.0%)	240 (48.1%)	249 (49.9%)		
Dead	10 (2.0%)	9 (1.8%)	1 (0.2%)		

**Table 4 ijms-25-00489-t004:** Univariate and multivariate analysis of the correlation of GLIS1 expression with OS among prostate cancer patients.

Characteristics	Total (N)	Univariate Analysis		Multivariate Analysis	
		Hazard Ratio (95% CI)	*p* Value	Hazard Ratio (95% CI)	*p* Value
Age	499				
≤60	224	Reference			
>60	275	1.577 (0.440–5.648)	0.484		
T stage	492				
T2	189	Reference			
T3 & T4	303	3.294 (0.612–17.727)	0.165		
N stage	426				
N0	347	Reference			
N1	79	3.516 (0.778–15.896)	0.102		
PSA (ng/mL)	442				
<4	415	Reference			
≥4	27	10.479 (2.471–44.437)	0.001	3.411 (0.735–15.826)	0.117
Gleason score	499				
6 & 7	293	Reference			
8 & 9 & 10	206	6.664 (1.373–32.340)	0.019	5.372 (0.843–34.209)	0.075
Residual tumor	468				
R0	315	Reference			
R1 & R2	153	2.598 (0.696–9.694)	0.155		
GLIS1	499				
Low	249	Reference			
High	250	0.122 (0.015–0.965)	0.046	0.157 (0.018–1.355)	0.092

**Table 5 ijms-25-00489-t005:** Association between GLIS1 and prostate cancer clinicopathological features.

Characteristics	Total (N)	Odds Ratio (OR)	*p* Value
Age (>60 vs. ≤60)	499	1.007 (0.708–1.434)	0.968
T stage (T3 & T4 vs. T2)	492	0.585 (0.404–0.843)	0.004
N stage (N1 vs. N0)	426	0.521 (0.311–0.859)	0.012
PSA (ng/mL) (≥4 vs. <4)	442	0.316 (0.122–0.731)	0.011
Gleason score (8 & 9 & 10 vs. 6 & 7)	499	0.645 (0.450–0.922)	0.017
Race (Black or African American & White vs. Asian)	484	3.130 (0.921–14.237)	0.090
Residual tumor (R1 & R2 vs. R0)	468	0.671 (0.454–0.990)	0.045

**Table 6 ijms-25-00489-t006:** Correlation analysis between GLIS1 and markers of immune cells in PRAD by TIMER.

	Gene Markers	PRAD			
		None		Purity	
		Correlation	*p*	Correlation	*p*
CD8+ T cell	CD8A	0.423	4.54 × 10^−23^	0.238	9.38 × 10^−7^
	CD8B	0.275	4.27 × 10^−10^	0.158	1.24 × 10^−3^
T cell (general)	CD3D	0.329	4.71 × 10^−14^	0.115	1.91 × 10^−2^
	CD3E	0.405	4.8 × 10^−21^	0.203	2.98 × 10^−5^
	CD2	0.366	2.94 × 10^−17^	0.181	2.15 × 10^−4^
B cell	CD19	0.247	2.42 × 10^−8^	0.11	2.45 × 10^−2^
	CD79A	0.221	6.67 × 10^−7^	0.077	1.18 × 10^−1^
Monocyte	CD86	0.284	1.00 × 10^−10^	0.131	7.67 × 10^−3^
	CD115 (CSF1R)	0.481	3.08 × 10^−30^	0.344	5.46 × 10^−13^
TAM	CCL2	0.302	5.37 × 10^−12^	0.149	2.38 × 10^−3^
	CD68	−0.311	7.92 × 10^−11^	0.08	1.04 × 10^−1^
	IL10	0.29	4.42 × 10^−11^	0.188	1.17 × 10^−4^
M1 macrophage	INOS (NOS2)	0.29	4.42 × 10^−11^	0.188	1.17 × 10^−4^
	IRF5	0.133	2.95 × 10^−3^	0.064	1.91 × 10^−1^
	COX2 (PTGS2)	0.353	4.81 × 10^−16^	0.254	1.46 × 10^−7^
M2 macrophage	CD163	0.185	3.15 × 10^−5^	0.061	2.15 × 10^−1^
	VSIG4	0.29	4.12 × 10^−11^	0.159	1.18 × 10^−3^
	MS4A4A	0.224	4.57 × 10^−7^	0.082	9.54 × 10^−2^
Neutrophils	CD66b (CEACAM8)	0.038	3.99 × 10^−1^	0.007	8.79 × 10^−1^
	CD11B (ITGAM)	0.387	3.08 × 10^−19^	0.241	6.59 × 10^−7^
	CCR7	0.306	2.97 × 10^−12^	0.089	7.11 × 10^−2^
Natural killer cell	KIR2DL1	0.046	3.02 × 10^−1^	−0.048	3.25 × 10^−1^
	KIR2DL3	0.118	8.44 × 10^−3^	0.115	1.89 × 10^−2^
	KIR2DL4	0.116	9.66 × 10^−3^	0.007	8.82 × 10^−1^
	KIR3DL1	0.101	2.44 × 10^−2^	−0.006	8.96 × 10^−1^
	KIR3DL2	0.094	3.54 × 10^−2^	0.055	2.65 × 10^−1^
	KIR3DL3	0.004	9.35 × 10^−1^	0.063	1.99 × 10^−1^
	KIR2DS4	0.148	9.07 × 10^−4^	0.092	5.98 × 10^−2^
Dendritic cell	HLA-DPB1	0.49	1.75 × 10^−31^	0.326	9.84 × 10^−12^
	HLA-DQB1	0.264	2.08 × 10^−9^	0.132	6.87 × 10^−3^
	HLA-DRA	0.351	7.66 × 10^−16^	0.161	1.00 × 10^−3^
	HLA-DPA1	0.423	4.35 × 10^−23^	0.271	1.93 × 10^−8^
	BDCA-1 (CD1C)	0.442	3.4 × 10^−25^	0.268	2.70 × 10^−8^
	BDCA-4 (NRP1)	−0.004	9.27 × 10^−1^	−0.023	6.44 × 10^−1^
	CD11c (ITGAX)	0.244	3.67 × 10^−8^	0.087	7.61 × 10^−2^
Th1	T-bet (TBX21)	0.332	2.9 × 10^−14^	0.154	1.63 × 10^−3^
	STAT4	0.354	3.94 × 10^−16^	0.178	2.56 × 10^−4^
	STAT1	0.087	5.35 × 10^−2^	−0.01	8.43 × 10^−1^
	IFN-γ (IFNG)	0.091	4.19 × 10^−2^	−0.002	9.75 × 10^−1^
	TNF-α (TNF)	0.214	1.52 × 10^−6^	0.096	4.92 × 10^−2^
Th2	GATA3	0.643	1.62 × 10^−59^	0.545	1.33 × 10^−33^
	STAT6	0.437	1.13 × 10^−24^	0.356	7.61 × 10^−14^
	STAT5A	0.622	1.33 × 10^−54^	0.499	1.30 × 10^−27^
	IL13	0.058	1.95 × 10^−1^	0.014	7.79 × 10^−1^
Tfh	BCL6	0.288	6.22 × 10^−11^	0.146	2.90 × 10^−3^
	IL21	0.053	2.39 × 10^−1^	0.008	8.73 × 10^−1^
Th17	STAT3	0.255	7.41 × 10^−9^	0.148	2.53 × 10^−3^
	IL17A	0.159	3.56 × 10^−4^	0.023	6.47 × 10^−1^
Treg	FOXP3	0.188	2.41 × 10^−5^	0.075	1.27 × 10^−1^
	CCR8	0.125	5.06 × 10^−3^	0.025	6.11 × 10^−1^
	STAT5B	0.5	6.45 × 10^−33^	0.432	2.33 × 10^−20^
	TGF-β (TGFB1)	0.532	1.08 × 10^−37^	0.412	1.94 × 10^−18^
T cell exhaustion	PD-1 (PDCD1)	0.317	4.27 × 10^−13^	0.128	8.92 × 10^−3^
	CTLA4	0.146	1.08 × 10^−3^	−0.025	6.17 × 10^−1^
	TIM-3 (HAVCR2)	0.275	4.25 × 10^−10^	0.122	1.30 × 10^−2^
	GZMB	0.222	5.57 × 10^−7^	0.023	6.37 × 10^−1^
	LAG3	0.466	3.62 × 10^−28^	0.346	3.67 × 10^−13^
	PDL1 (CD274)	0.286	7.63 × 10^−11^	0.158	1.23× 10^−3^

TAM, tumor-associated macrophage; Th, T helper cell; Tfh, Follicular helper T cell; Treg, regulatory T cell; Cor, R value of Spearman’s correlation; None, correlation without adjustment; Purity, correlation adjusted by purity.

## Data Availability

miRTarBase, http://miRTarBase.cuhk.edu.cn; Targetscan, https://www.targetscan.org; miRWalk, http://mirwalk.umm.uni-heidelberg.de; ENCORI, https://starbase.sysu.edu.cn; GEO database, http://www.ncbi.nlm.nih.gov/geo; GTEx database, https://gtexportal.org; TCGA database, https://portal.gdc.cancer.gov; TIMER, https://cistrome.shinyapps.io/timer.

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
