# Peer review of "GLIS1, Correlated with Immune Infiltrates, Is a Potential Prognostic Biomarker in Prostate Cancer"

_ijms, 2023, doi:10.3390/ijms25010489_

Round 1

Reviewer 1 Report

Comments and Suggestions for Authors

In this manuscript, the authors study the role of GLIS1 in prostate cancer. The study uses different databases and highlights how GLIS1 can be a prognostic factor in this cancer. Furthermore, they correlate the GLIS1 expression with the immune infiltrate; for this purpose, it would be interesting to study the correlation between GLIS1 and PDL1 expression. The manuscript is clear and well-written, but the data should be validated using human samples or cell systems. For this reason, this journal is not appropriate for this kind of study. 

A bioinformatic journal would be better.. 

Reviewer 2 Report

Comments and Suggestions for Authors

Your work underlines the need of  exploration of genetic landascape of prostate cancer and relationship with inflammatory tumoral response and other features as tumor micro-environment.

High quality genetical investigation and use of large datasets enhanced knowledge of a burden of unkown factors contribuiting to oncogenesis and cancer progression

THis work enhances the need of new well designed trials focusing on biomarker/genetical ordinary analysis and correct combination of therapheutical approaches.

This work has relevant impact on knowledge of initiation and progression of PC, starting from miRNA going inside to a number of plausible factors. 

We have to take ispiration from other malignancies as ovarian and breast cancer in wich tumor profiling and genetical investigations are actually key points driving terapies and trials every day more focused.

You could add some other clinical perspective to integrate this kind of results in new well designed trials.

Reviewer 3 Report

Comments and Suggestions for Authors

The article on "GLIS1, Correlated with Immune Infiltrates, Is a Potential Prognostic Biomarker in Prostate Cancer" offers valuable insights into miRNA, specifically GLIS1, in prostate cancer. To enhance impact and accessibility, consider streamlining the introduction for succinctness while retaining key information and reinforcing statistical data.

In the methods section, the methodology for identifying DEGs and DEMs in PCa using GEO datasets is well-documented. However, provide more clarity on the rationale behind criteria for significant DEGs and DEMs to enhance understanding.

The results section effectively explores DEGs and DEMs, focusing on GLIS1, with integration of GO analysis and KEGG pathways. To improve comprehension, incorporate visual aids such as figures or tables illustrating complex relationships and findings.

The discussion delves into GLIS1's implications in PCa, emphasizing its potential as an immune-related biomarker. Strengthen the discussion by explicitly addressing potential biases or limitations.

In the conclusion, briefly summarize key findings and underscore GLIS1's significance in PCa. Consider a slight expansion on broader implications for future research or clinical applications to provide a more comprehensive closure.

Comments on the Quality of English Language

The English language quality is good 

Reviewer 4 Report

Comments and Suggestions for Authors

The authors present an analysis of GLIS1, and its correlation with immune infiltrates, and address whether it might be a suitable prognostic marker for prostate cancer. The document is generally well-written and the data science appears sound.

The authors make a very strong statement in that GLIS1 correlates with all factors of better prognosis and long survival, such as lower Gleason score and PSA. I am not convinced that GLIS1 correlates with the initiation and progression of prostate cancer; it seems to me a better marker of indolent disease. The authors need to compare GLIS1 ‘high’ and ‘low’ expression in a large cohort with different NCCN risk categories and check if GLIS1 is an independent predictor or better than the current most used option to estimate risk.

It would be intriguing to assess GLIS expression staining on cohorts using immunotherapy (IO) in prostate cancer. As far as I am aware, the prostate is an immune-desert cancer, but perhaps this marker could identify the responders. In parallel, I would test the hypothesis of IO in xenografts that are GLIS ‘low’ versus ‘high’. Again, there is an inconsistency here with clinical data – indolent tumors, that seem to be GLIS high, are exactly the ones with poor response to immunotherapy. Thus not as clear cut as the paper suggests.

There are a few errors of grammar and the manuscript might benefit from an editing service. Some examples:

Line 42, "Immunotherapy, a promising strategy that has shown antitumor effects in PCa [8]." should probably read "Immunotherapy is a promising strategy that has shown antitumor effects in PCa [8].

Line 60 and other instances, "Regarding tumors, GLIS1 in cancer cells was involved in cell migration, invasion and tumorigenesis [21, 22]." Presumably, we still believe this to be the case, so the present tense is more correct "Regarding tumors, GLIS1 in cancer cells is involved in cell migration, invasion and tumorigenesis [21, 22]."

Several of the figures could be better presented, for example:

Figure 6: parts A and B leave a lot of spare white space, whilst part C is too small, and would benefit from being e.g. over two rows of faceted panels. Please ensure that Figures use their space wisely to ensure maximum readability.

Finally, some of the statistical work may be confusing, especially to the non-expert reader, for example:

Line 525, it is not clear why the authors have used different fold changes (1.2 and 1.5) for DEGs and DEMs. The non-expert reader will not understand this distinction, please provide a reference explaining why 'best practice' thresholds for fold change should be different.

Line 559, we now move to a log2 fold change >1 as a cut-off, again, non-expert readers will be confused as to why p-value < 0 .05 is consistently used, but fold changes are used with a variety of thresholds. Ideally, the same statistical threshold should be used throughout, or a strong justification should be provided. Please be very careful here in terms of data presentation.

Line 591, the authors use Wilcoxon rank-sum test and Wilcoxon signed-rank test. It would be worth explaining that this is (presumably) due to the expression of GLIS1 not being normal - the non-expert reader may wonder whether the Kolmogorov-Smirnov Test or similar was employed, or any transformations to convert the data (e.g. log-normal) investigated? It is helpful for reproducibility to explain why tests were performed; not just what tests were done.

Smaller points for correction:

Line 602, "Supplementary Materials: The following supporting information can be downloaded at:" It would be helpful to the reader to see a summary of what figures and tables are in the supporting information.

Line 612, "Data Availability Statement: Not applicable." This is unhelpful. The data used in this work are available and this section could easily be adjusted to provide guidance to the reader (URLs etc) as to precisely where the data sources came from.

Line 640, something appears to have gone wrong with the formatting of Reference 1.

Comments on the Quality of English Language

As noted above there needs proof reading and syntax, grammar correction.

Round 2

Reviewer 4 Report

Comments and Suggestions for Authors

Thankyou for your reply. V helpful. 

Comments on the Quality of English Language

It’s ok.